EMBO
Molecular Medicine

# Olverembatinib inhibits SARS-CoV-2-Omicron variant-mediated cytokine release in human peripheral blood mononuclear cells

Marina Chan[1], Eric C Holland[1] & Taranjit S Gujral[1,2,*] 

The World Health Organization has declared COVID-19 to be a pandemic. Despite the development of vaccines, COVID-19 continues to be a healthcare burden, especially in persons with a compromised immune system and others who remain unvaccinated. Most COVID-19 patients develop mild-to-moderate symptoms, while 15–20% of patients face hyper-inflammation induced by massive cytokine production, called "cytokine storm," ultimately leading to alveolar damage and respiratory failure. Recently, we and others have shown that stimulation with S1 subunit of SARS-CoV-2 spike protein causes upregulation and release of a panel of inflammatory molecules such as IL-1b, IL-6, and CCL7 in monocytes and peripheral blood mononuclear cells (PBMCs) (Chan *et al*, 2021; Kucia *et al*, 2021; Lu *et al*, 2021). Further, it has been shown that stimulation with the N-terminus domain (NTD) of the S1 subunit is sufficient to activate monocytes. Consistently, recent studies have identified several C-type lectins and Tweety family member 2 as glycan-dependent binding partners of the NTD of the SARS-CoV-2 spike protein (Lempp *et al*, 2021; Lu *et al*, 2021). The engagement of these receptors with the SARS-CoV-2 virus induces robust proinflammatory responses in myeloid cells that correlate with COVID-19 severity. Thus, identifying potential therapeutics that could abrogate NTD-mediated activation of monocytes and subsequent cytokine release is highly desirable for treating moderate to

severe COVID-19 and other conditions where a cytokine storm is a lethal event.

In late November 2021, the Omicron (B.1.1.529 / 21K) variant was detected in South Africa and has been associated with rapidly increasing case numbers worldwide (preprint: Pulliam *et al*, 2021). The Omicron variant carries more than 50 mutations, including more than 25 mutations in the Spike protein alone. Of these, ~30% of mutations are present in the NTD (Fig 1A). Here, we sought to determine whether the mutations in the NTD of the Omicron variant affect its ability to activate myeloid cells and promote the secretion of inflammatory cytokines. First, we stimulated pooled PBMCs with mammalian cell (HEK293)-derived NTD (1 μg/ml) of the Omicron variant. PBMCs from healthy donors spanning various age groups were obtained from Bloodworks NW, Seattle, Washington. We observed a robust (> 100-fold) increase in the release of interleukins including IL-1β, IL-6, and tumor necrosis factor (TNFα) (Fig 1B), suggesting that the Omicron variant NTD can still activate myeloid cells and promote cytokine release. Next, we compared the changes in cytokine release in PBMCs produced by the stimulation of NTD from the Omicron, Delta, and Wuhan variants. Recombinant purified NTD of the SARS-CoV-2 Wuhan, Delta, and Omicron variants were obtained from Acro Biosystems. Our data show that the NTD from the Omicron variant was equally effective in promoting cytokine release (Fig 1C).

Together, these data establish that the presence of mutations in the NTD of the Omicron variant does not alter its ability to promote cytokine release.

Previously, a machine learning-based drug screening identified Ponatinib, an FDA-approved drug for chronic myelogenous leukemia (CML), as a potent inhibitor of S1 protein-mediated cytokine release in PBMCs. (Chan *et al*, 2021). Thus, we asked whether Ponatinib treatment could also inhibit cytokine release mediated by the NTD from the Omicron variant. In addition, we also evaluated the efficacy of Baricitinib, an FDA-approved JAK inhibitor for the treatment of COVID-19 (Favalli *et al*, 2020), and Olverembatinib, a clinical-stage multi-kinase inhibitor that is structurally similar to ponatinib, with a manageable safety profile (Jiang *et al*, 2019). Olverembatinib is being marketed in China for treating resistant CML patients with T315I mutation by the National Medical Products Administration in China, and in phase Ib clinical trials in the US for Ponatinib resistant and/or intolerant CML patients (Dhillon, 2022). Our data show that treatment with Ponatinib or Olverembatinib inhibited the NTD-mediated release of all seven cytokines measured in a dose-dependent manner (Fig 2A). Both Ponatinib and Olverembatinib inhibited the release of cytokines even at low nanomolar concentrations (< 50 nM). In contrast, treatment with Baricitinib only inhibited three of seven cytokines measured (GM-CSF, CCL2, and IL-10) at 1,000 nM (Fig 2A). Since

---

1 Human Biology Division, Fred Hutchinson Cancer Center, Seattle, WA, USA
2 Department of Pharmacology, University of Washington, Seattle, WA, USA
*Corresponding author. E-mail: tgujral@fredhutch.org

DOI 10.15252/emmm.202215919 | EMBO Mol Med (2022) 14: e15919 | Published online 17 May 2022

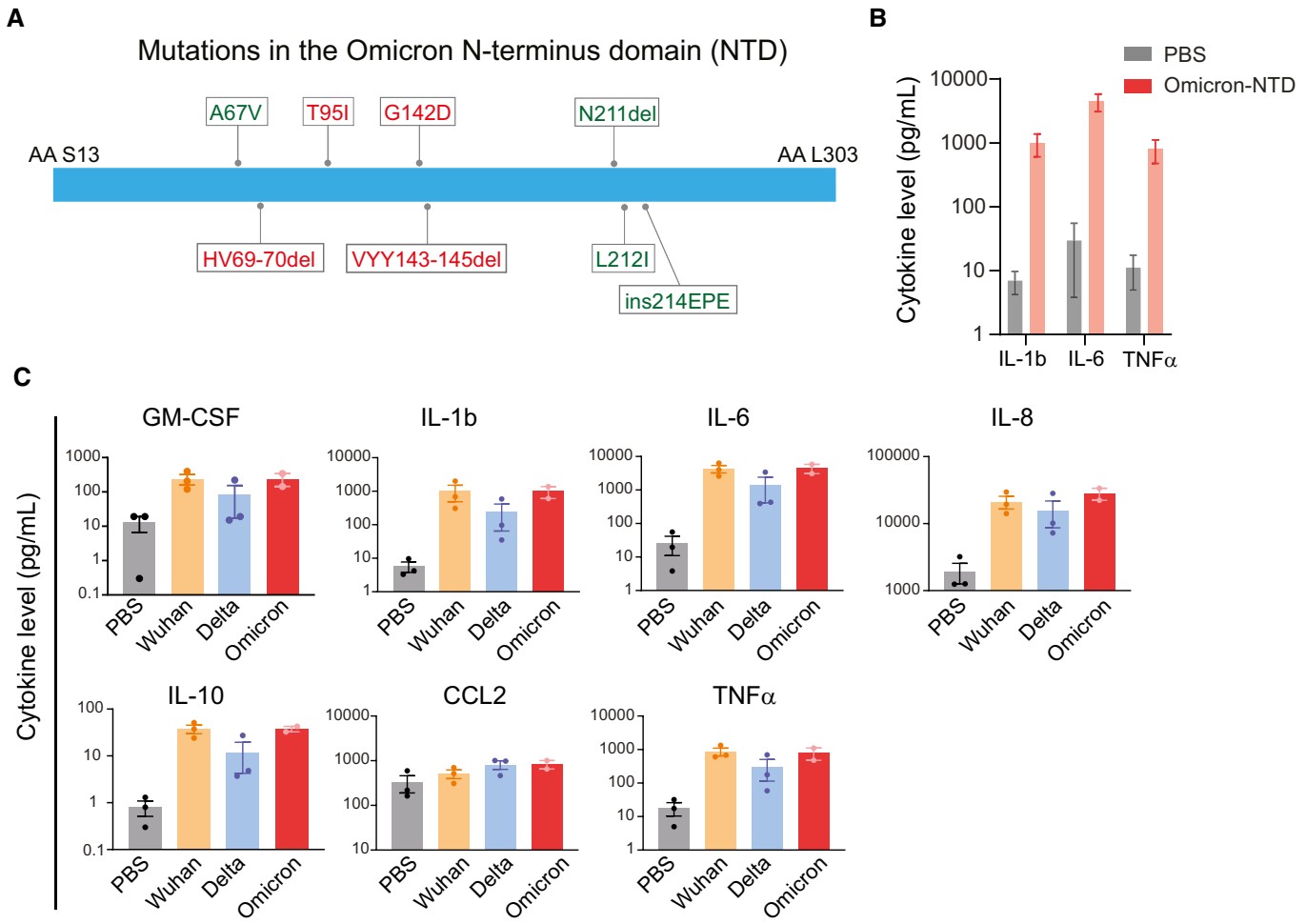

**Figure 1. NTD of the SARS-CoV-2 Omicron variant stimulates cytokine release.**

(A) A schematic showing mutation in the Omicron NTD. Unique mutations found in the Omicron variant are shown in green color. (B) Changes in cytokine release in response to the recombinant NTD from Omicron variant. Measurement of cytokine release from healthy donor PBMCs treated with PBS or Omicron NTD at 1 μg/ml for 24 h. Cytokines were measured by Luminex multiplex assay. (C) Comparison of cytokine release from healthy donor PBMCs treated with Wuhan, Delta, or Omicron NTD at 1 μg/ml for 24 h. Cytokine release in the conditioned media was measured by Luminex. Recombinant NTD of different variants were purified from HEK293 cells. Data are shown as the mean of two to three biological replicates. Error bars denote SEM.

Olverembatinib treatment showed the most substantial suppression of the Omicron-NTD-mediated cytokine release, we next determined the Olverembatinib concentration range for the panel of cytokines tested, required to inhibit 50% of the Omicron NTD-mediated cytokine release ($EC_{50}$), was between 7.7 and 56 nM (Fig 2B). Previously, several protein kinases, including JAK1, EPHA7, IRAK1, MAPK12, and MAP3K3, were identified as essential for S1 protein-mediated cytokine release in myeloid cells (Chan *et al*, 2021). The kinase activity profile of Olverembatinib showed that this drug inhibits 11 of 13 kinases predicted to be essential for the NTD-mediated chemokine and cytokine release (Fig 2C). Thus, these data indicate that Olverembatinib, a clinical-grade, multi-specific kinase inhibitor, blocks the activity of several kinases essential for cytokine signaling, thereby dampening the Omicron NTD-mediated cytokine release. Finally, we evaluated the response of Olverembatinib in PBMCs from nine COVID19 patients. Consenting SARS-CoV-2-infected (*n* = 9) donors, age 18 years and older, provided anticoagulated blood samples by venipuncture at the Seattle Vaccine Trials Unit. Our data show that Omicron variant-NTD stimulation (1ug/mL) causes a significant increase in the release of all five cytokines measured in the conditioned media from COVID19 PBMCs (Fig 2D). Interestingly, a subset of cytokines and chemokines showed a large variable response to Omicron variant-NTD, suggesting an inherent patient-to-patient variability. However, treatment with 100 nM Olverembatinib completely shuts down NTD-mediated cytokine storm in all COVID-19 PBMCs.

Taken together, we propose that agents targeting multiple kinases essential for SARS-CoV-2-mediated cytokine release, such as Olverembatinib and Ponatinib, may represent an attractive therapeutic option for treating moderate-to-severe COVID-19.

**Expanded View** for this article is available online.

## Acknowledgements

This work was supported by grants from the Fred Hutch COVID-19 Pilot Fund. Fred Hutchinson Cancer Center Institutional Review Board approved all

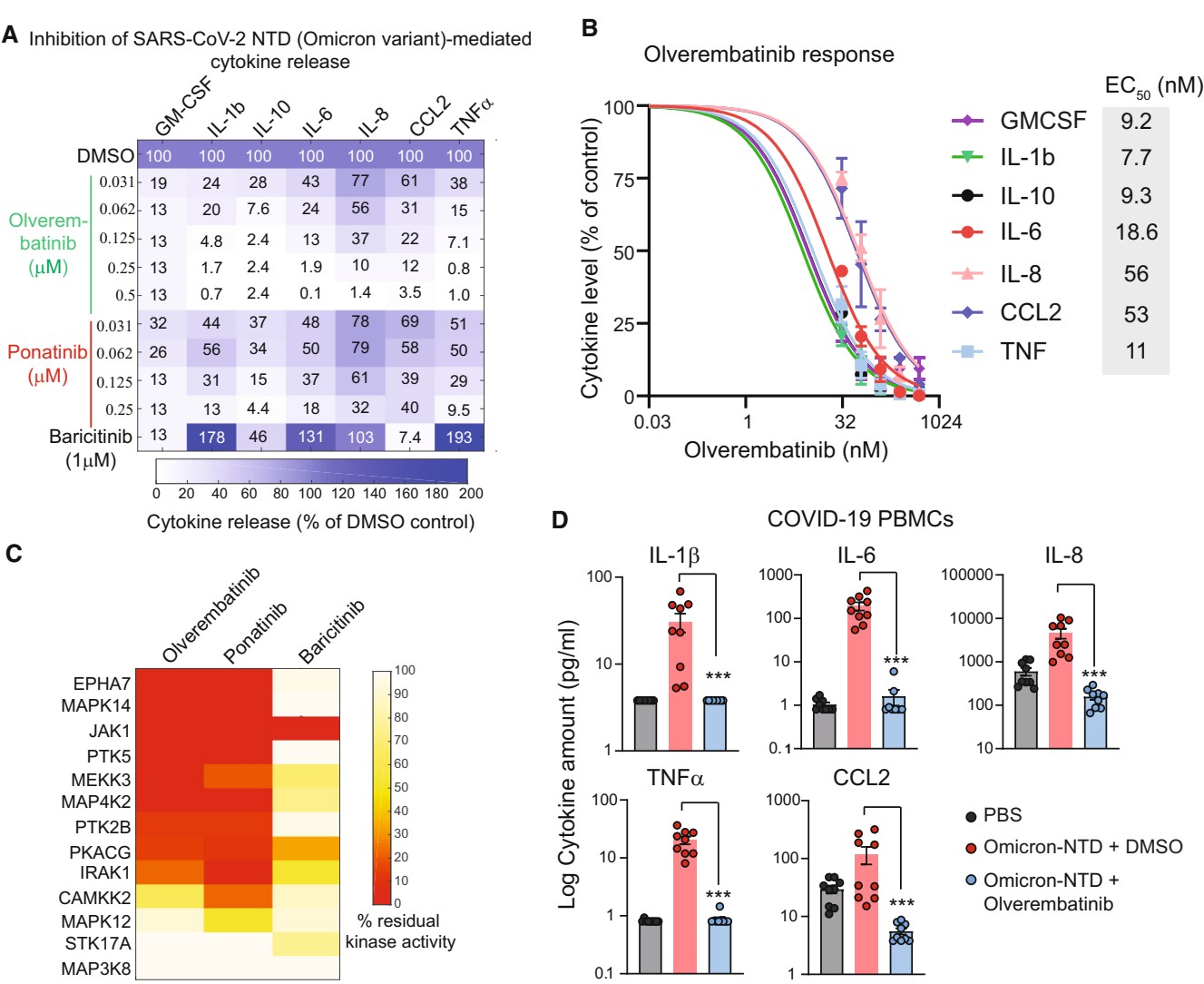

**Figure 2. Olverembatinib is a potent inhibitor of Omicron NTD-mediated cytokine release.**

(A) Effect of Olverembatinib, Ponatinib, and Baricitinib on Omicron NTD-mediated cytokine release. (B) EC50 of Olverembatinib on indicated cytokines. (C) Comparison of kinase inhibition profiles of Olverembatinib, Ponatinib, and Baricitinib. (D) Olverembatinib treatment (100 nM) decreases Omicron-variant-NTD-mediated cytokine release in COVID-19 PBMCs. Plots showing changes in indicated cytokines in response to Omicron variant-NTD at 1 μg/ml and Olverembatinib in COVID-19 PBMCs for 24 h. Data are the mean of nine biological replicates, and error bars indicate SEM. *** represent $P < 0.003$, Mann–Whitney $U$-test.

aspects of this study (IRB 10440, 00001080, and 00022371). Informed Consent was obtained from all subjects, and experiments conformed to the principles set out in the WMA Declaration of Helsinki and the Department of Health and Human Services Belmont Report. We thank Drs. Rachel Bender Ignacio, Vivian Oehler, and Josh Schiffer for helpful discussion on clinical implications of our findings. We also thank the staff at Immune Monitoring Core for their help and support throughout this project.

## Author contributions

**Marina Chan:** Conceptualization; Data curation; Investigation; Methodology; Writing—original draft. **Eric C Holland:** Formal analysis; Supervision; Writing—review & editing. **Taranjit S Gujral:** Conceptualization; Supervision; Investigation; Writing—review & editing.

In addition to the CRediT author contributions listed above, the contributions in detail are: MC, ECH, and TSG conceived the study. MC and TSG designed the study and performed experiments. MC, ECH, and TSG interpreted and discussed the data. MC wrote the draft with input from all authors.

## Disclosure statement and competing interests

The authors declare that they have no conflict of interest.

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
