## [Review Process File · EMBO Molecular Medicine]

Olverembatinib inhibits SARS-CoV-2-Omicron variant-mediated cytokine release in human peripheral blood mononuclear cells

Marina Chan, Eric Holland, and Taran Gujral
DOI: 10.15252/emmm.202215919

Corresponding author: Taran Gujral (tgujral@fredhutch.org)

Review Timeline:

Submission Date:	19th Feb 22
Editorial Decision:	14th Mar 22
Revision Received:	23rd Mar 22
Editorial Decision:	28th Mar 22
Revision Received:	7th Apr 22
Editorial Decision:	12th Apr 22
Revision Received:	13th Apr 22
Accepted:	19th Apr 22

Editor: Zeljko Durdevic

Transaction Report:

14th Mar 2022

Decision on your manuscript EMM-2022-15919

Dear Dr. Gujral,

Thank you for the submission of your manuscript to EMBO Molecular Medicine. We have now received feedback from the two reviewers who agreed to evaluate your manuscript.

As you will see from the reports pasted below, both referees raise serious concerns particularly regarding the overall conclusiveness of the presented data. As clear and conclusive insight into a novel, clinically relevant observation is crucial for publication in EMBO Molecular Medicine, and together with the fact that we only accept papers that receive enthusiastic support upon initial review, I am afraid that we cannot offer to consider the manuscript further.

I am sorry that I can't be more positive on this occasion. Please rest assured that this is not a judgment of the quality or interest of your work but a decision based on appropriateness for EMBO Molecular Medicine.

Yours sincerely,

Zeljko Durdevic

***** Reviewer's comments *****

Referee #1 (Remarks for Author):

This is a very short piece, really a letter but with an abstract. The conclusions are relevant, but which kinases are involved? To establish this requires kinase assays versus speculation.

Referee #2 (Novelty/Model system Comments for Author):

The authors found that NTD of SARS-CoV-2 spike protein could stimulate PBMCs to secrete cytokines, and agents such as Olverembatinib and Ponatinib could reduce the cytokine release via blocking relative kinase activity. It's interesting that these drugs might provide a therapeutic option for treating COVID-19 patients. However, the limited samples (probably three?) and lack of physiological experiments limit the credibility of their conclusions.

Referee #2 (Remarks for Author):

The authors found that NTD of SARS-CoV-2 spike protein could stimulate PBMCs to secrete cytokines, and agents such as Olverembatinib and Ponatinib could reduce the cytokine release via blocking relative kinase activity. It's interesting that these drugs might provide a therapeutic option for treating COVID-19 patients. However, the limited samples (probably three?) and lack of physiological experiments limit the credibility of their conclusions.

Major questions:

1. They mentioned that "We and others have recently reported that the N-terminus domain (NTD) of the SARS-CoV-2 of various variants is responsible for inducing cytokine release in human PBMCs." Do the authors mean N-terminus domain (NTD) of SARS-CoV-2 spike protein? If so, please rephrase them.
2. It seems that the authors performed the experiment with PBMCs from three healthy donors. The cohort was too small to support such conclusions. In addition, the inflammatory status of PBMCs in COVID-19 patients were quite different from that in healthy people. Do Olverembatinib and other drugs also inhibit NTD-mediated cytokine release of PBMCs from COVID-19 patients?
3. Could the drugs mentioned in the manuscript inhibit the activity of relative kinases in mouse or other animal models? If so, the authors should verify the pesticide effect in animal models such as hACE2-transgenic mice which infected with SARS-CoV-2 virus. If these candidates could reduce cytokine release in vivo, the authors may find out more detail mechanisms of cytokine

inhibition in immune cells and confirm their conclusion.

4. Lack of Materials and Methods.

As a service to authors, EMBO provides authors with the possibility to transfer a manuscript that one journal cannot offer to publish to another EMBO publication. The full manuscript and if applicable, reviewers reports are automatically sent to the receiving journal to allow for fast handling and a prompt decision on your manuscript. For more details of this service, and to transfer your manuscript to another EMBO title please click on Link Not Available

Dear Zeljko,

Thank you for the timely handling of our manuscript. I fully understand that you are prepared to reject the paper with some reservations by the second reviewer. However, I believe that the reservations by reviewers could be handled by textual clarifications, clarification of the formatting of a correspondence article, and additional data that we have already generated.

Firstly, we think both reviewers were unaware of our recent manuscript published in Mol Sys Bio<<https://www.embopress.org/doi/full/10.15252/msb.202110426>> and an independent study published in Immunity<[Secondly, we have already generated data on the efficacy of Olverembatinib on Omicron-NTD mediated cytokine release in PBMC from COVID19 patients \(as suggested by reviewer 2\). Therefore, we can provide these data in the revised manuscript. I have attached a document describing our point-by-point response to each reviewer's concerns.](https://urldefense.proofpoint.com/v2/url?u=https-3A__www.sciencedirect.com_science_article_pii_S1074761321002120&d=DwMFaQ&c=eRAMFD45gAfqt84VtBcfhQ&r=bNOSf7W5PfZ1u6Fq_DvVDrtPKrLkQXfJv471uCe58Cg&m=z51ZmFeyTeQxKg9J9jwwO47B3-VbNholadjlbnMsW4&s=bH4hCuCBXWjViAj2GJyqSCrmjAlBpQCggUdqtFV24GU&e=>, which showed that the SARS-CoV-2 virus induces cytokine production in human myeloid cells. Our previous work identified several new downstream kinases and effective drugs that could dampen NTD-mediated cytokine storm. In addition, this study published in Immunity replicated some of the findings in our manuscript and discovered new monocyte-specific receptors that recognize S1 spike protein to promote cytokine production. Frankly, it is not the reviewer's fault. The COVID19 research is moving at an unprecedented speed, and no one can keep track of new developments. Our new manuscript, a correspondence, provides new and timely data on NTD from the Omicron variant and includes data that a clinical-grade compound called Olverembatinib could inhibit Omicron-mediated cytokine release. Our clinical team and we are excited by these findings and have already initiated a Phase II clinical trial to test Olverembatinib in COVID-19 patients (see attached clinical protocol).

I look forward to receiving your revised manuscript.

Yours sincerely,

Zeljko Durdevic

We are grateful to the reviewers for their thoughtful suggestions and comments. We will address each of their specific concerns individually

Reviewer #1:

1. This is a very short piece, really a letter but with an abstract. The conclusions are relevant, but which kinases are involved? To establish this requires kinase assays versus speculation.

We thank the reviewer for sharing our enthusiasm for these findings. In our recent manuscript published in Molecular Systems Biology, we identified and validated a set of seven kinases (MAPK12, EPHA7, MAP3K8, PRKACG, IRAK1, MAP3K3, and JAK1) that play an important role in the NTD-mediated cytokine release (see **Figure 1 below**).

In this short manuscript (a correspondence), we present new and timely data showing a clinical-grade inhibitor Olverembatinib, that targets **five out of seven** kinases and inhibits Omicron variant-NTD-mediated cytokine release (**Figure 2 in the manuscript**).

Figure 1. Functional screening identified key kinase drivers of the NTD stimulated cytokine release. Validation of predicted kinases as drivers of cytokine release by siRNAs in THP1 cells. Cytokine release was measured by Luminex and normalized to

Reviewer #2:

1. They mentioned that "We and others have recently reported that the N-terminus domain (NTD) of the SARS-CoV-2 of various variants is responsible for inducing cytokine release in human PBMCs." Do the authors mean N-terminus domain (NTD) of SARS-CoV-2 spike protein? If so, please rephrase them.

In the revised manuscript, we will have rephrased this sentence.

2. It seems that the authors performed the experiment with PBMCs from three healthy donors. The cohort was too small to support such conclusions. In addition, the inflammatory status of PBMCs in COVID-19 patients were quite different from that in healthy people. Do Olverembatinib and other drugs also inhibit NTD-mediated cytokine release of PBMCs from COVID-19 patients?

In the revised manuscript, we present new data from PBMCs derived from COVID-19 patients (see **new Figure 2D** and results section on **page 4**) . As pointed out by the reviewer, there is higher variability in terms of NTD-mediated changes in cytokines in COVID-19 PBMCs (See **New Figure 2D**) . Our new data from COVID-19 PBMCs further supports our findings that Olverembatinib potently inhibits NTD-mediated cytokine release of PBMCs from COVID-19 patients.

3. Could the drugs mentioned in the manuscript inhibit the activity of relative kinases in mouse or other animal models? If so, the authors should verify the pesticide effect in animal models such as hACE2-transgenic mice which infected with SARS-CoV-2 virus. If these candidates could reduce cytokine release in vivo, the authors may find out more detail mechanisms of cytokine inhibition in immune cells and confirm their conclusion.

Unfortunately, due to the safety requirements of the ABSL-3 standard, we are unable to perform direct testing of drugs on SARS-CoV-2-mediated cytokine storm in the animal model. However, in the recently published paper, we evaluated the efficacy of Ponatinib in lipopolysaccharide (LPS)-induced pulmonary inflammation model in mice.

The LPS-induced models of Acute lung injury (ALI) and acute respiratory distress syndrome (ARDS) in mice are well-established in vivo models to study pulmonary infection. Further, both ALI and ARDS are known to occur in the clinical presentation of severe SARS-CoV-2 disease (Li et al, 2020). Thus, we sought to compare S1 protein and LPS-mediated changes in cytokine expression in THP1 cells and determine whether Ponatinib could inhibit LPS-mediated cytokine production in these cells. Our data show that both S1 spike protein (1mg/mL) and LPS (1mg/mL) stimulation increased the expression of all measured cytokines (**see Figure 2 below**). Consistent with these in vitro data, we show that a 1-hour pre-treatment with Ponatinib (35 mg/Kg) significantly reduces symptoms of acute lung inflammation in the LPS-induced lung inflammation mouse model (**Figure 2 below**). Treatment with Ponatinib at 35 mg/kg alleviated LPS-induced lung injury, including interstitial and intra-alveolar edema, septal thickening, alveolar collapse, and inflammatory cell infiltration, assessed by histology (**Figure 2 below**). Bronchoalveolar lavage fluid (BALF) collected at 5 hours post LPS treatment showed a significant reduction in GM-CSF, IL-6, and TNF α levels measured

by Luminex (Figure 2 below). Together, these data suggest that Ponatinib also blocks LPS-mediated downstream signaling and cytokine production.

Figure. 2. Ponatinib alleviates symptoms of acute lung inflammation in LPS-induced lung inflammation mouse model. **A.** Comparison of LPS and full length S1 spike protein-mediated changes in cytokines in THP1 macrophages. **B.** Ponatinib inhibits LPS-mediated cytokine release in THP1 macrophages *in vitro*. **C.** A schematic showing the overall design of *in vivo* study evaluating the efficacy of ponatinib in LPS-induced lung inflammation mouse model. **D.** Representative H&E images showing Ponatinib alleviates LPS-induced inflammatory cell infiltration, septal thickening, alveolar edema in mouse lungs.

4. Lack of Materials and Methods.

In the revised manuscript, we have added detailed materials and methods in the supplementary information section

12th Apr 2022

Dear Dr. Gujral,

Thank you for the submission of your manuscript to EMBO Molecular Medicine. I am pleased to inform you that we will be able to accept your manuscript pending the following final amendments:

- 1) Title: Title should include information about the model system i.e. "Olverembatinib inhibits SARS-CoV-2-Omicron variant-mediated cytokine release in human peripheral blood mononuclear cells".
- 2) Figures: Please upload individual, high-resolution figure files. Remove figures from the main manuscript file and leave figure legends at the end of the main manuscript text. For more information on figure presentation please check "Author Guidelines". <https://www.embopress.org/page/journal/17574684/authorguide#datapresentationformat>
- 3) In the main manuscript file, please do the following:
 - Add up to 5 keywords.
 - Add contributions for all authors.
 - Add "Disclosure Statement & Competing Interests". We updated our journal's competing interests policy in January 2022 and request authors to consider both actual and perceived competing interests. Please review the policy <https://www.embopress.org/competing-interests> and update your competing interests if necessary.
- 4) Please provide 2 sentences that summarise the main message of the article.
- 5) Appendix: Please move the M&M section to the Point-by-point response to the referees. One sentence explanation of the methods and materials used should be included in the manuscript. Hence, following information should be added at the appropriate places in the main manuscript text:
 - PBMCs from healthy donors spanning various age groups were obtained from Bloodworks NW, Seattle, Washington.
 - Recombinant purified N-terminal domain (NTD) of the SARS-CoV-2 Wuhan, Delta, and Omicron variants were obtained from Acro Biosystems.
 - Consenting SARS-CoV-2-infected (n=9) donors, age 18 years and older, provided anticoagulated blood samples by venipuncture at the Seattle Vaccine Trials Unit.
 - Fred Hutchinson Cancer Research Center Institutional Review Board approved all aspects of this study (IRB 10440, 00001080 and 00022371). Informed Consent was obtained from all subjects, and experiments conformed to the principles set out in the WMA Declaration of Helsinki and the Department of Health and Human Services Belmont Report.
 - Cytokines were measured by Luminex multiplex assay.
- 6) As part of the EMBO Publications transparent editorial process initiative (see our Editorial at <http://embomolmed.embopress.org/content/2/9/329>), EMBO Molecular Medicine will publish online a Review Process File (RPF) to accompany accepted manuscripts. This file will be published in conjunction with your paper and will include the anonymous referee reports, your point-by-point response and all pertinent correspondence relating to the manuscript. Let us know whether you agree with the publication of the RPF and as here, if you want to remove or not any figures from it prior to publication. Please note that the Authors checklist will be published at the end of the RPF.
- 7) Please provide a point-by-point letter INCLUDING my comments as well as the reviewer's reports and your detailed responses (as Word file).

I look forward to reading a new revised version of your manuscript as soon as possible.

Yours sincerely,

Zeljko Durdevic

The authors performed the requested editorial changes.

We are pleased to inform you that your manuscript is accepted for publication and is now being sent to our publisher to be included in the next available issue of EMBO Molecular Medicine.